# Spatiotemporal Characteristics of Neural Dynamics in Theta Oscillations Related to the Inhibition of Habitual Behavior

**DOI:** 10.3390/brainsci11030368

**Published:** 2021-03-13

**Authors:** Jae-Hwan Kang, Junsuk Kim, Yang Seok Cho, Sung-Phil Kim

**Affiliations:** 1AI Grand ICT Research Center, Dong-eui University, Busan 47340, Korea; jh.kang@deu.ac.kr; 2Department of Industrial ICT Engineering, Dong-eui University, Busan 47340, Korea; 3Department of Psychology, Korea University, Seoul 02841, Korea; yscho_psych@korea.ac.kr; 4Department of Biomedical Engineering, Ulsan National Institute of Science and Technology, Ulsan 44919, Korea

**Keywords:** electroencephalography, rock–paper–scissors task, habitual behaviors, frontal theta oscillations, phase synchronization

## Abstract

The human brain carries out cognitive control for the inhibition of habitual behaviors by suppressing some familiar but inappropriate behaviors instead of engaging specific goal-directed behavior flexibly in a given situation. To examine the characteristics of neural dynamics related to such inhibition of habitual behaviors, we used a modified rock–paper–scissors (RPS) task that consisted of a basic, a lose-, and a win-conditioned game. Spectral and phase synchrony analyses were conducted to examine the acquired electroencephalogram signals across the entire brain during all RPS tasks. Temporal variations in frontal theta power activities were directly in line with the stream of RPS procedures in accordance with the task conditions. The lose-conditioned RPS task gave rise to increases in the local frontal power and global phase-synchronized pairs of theta oscillations. The activation of the global phase-synchronized network preceded the activation of frontal theta power. These results demonstrate that the frontal regions play a pivotal role in the inhibition of habitual behaviors—stereotyped and ingrained stimulus–response mappings that have been established over time. This study suggests that frontal theta oscillations may be engaged during the cognitive inhibition of habitual behaviors and that these oscillations characterize the degree of cognitive load required to inhibit habitual behaviors.

## 1. Introduction

Human information processing deals with relationships among presented external stimuli, subsequent cognitive responses, and behavioral outcomes. Previously established stimulus–response (S–R) mappings are able to either facilitate or inhibit relevant cognitive processes. Especially, task switching or rule changing in the formed S-R mappings leads to cognitive interferences or conflicts and a “switching cost”, that is, the higher use of cognitive resources [1,2,3]. As an essential modulator of flexible goal-directed behavior [4], cognitive control refers to the ability to regulate, coordinate, and sequence thoughts and actions in accordance with internally maintained behavioral goals [5]. For decades, studies have investigated cognitive theories and neural mechanisms underlying cognitive control to manipulate a variety of interferences and conflicts within S-R mappings [6,7,8,9,10,11,12,13]. To date, many types of experimental paradigms have been used in traditional stimulus–response compatibility (SRC) tests. For example, the Stroop [14], Simon [15], go/no-go [16], and flanker [17] paradigms investigate distinct cognitive processes by generating a variety of combination conditions between S-R mappings. The well-known rock–paper–scissors (RPS) game is also regarded as a useful experimental paradigm not only in cognitive psychology and neuroscience but also in clinical research. Studies that use the RPS game as the main experimental paradigm often modify the classical RPS game to create different goal-directed conditions [18,19,20,21,22]. For example, participants might be instructed to intentionally lose, win, or come to a draw in the RPS games. These instructions generate cognitive interference that can be resolved to achieve a specific goal by adapting and changing behavior flexibly [18,19,23]. Compared to other SCR tests comprising inhibition processes, a modified RPS task engages in the inhibition of more prepotent, habitual, stereotyped, and ingrained behaviors specifically [19].

Due to the fact that high-level execution functions, such as working memory, decision making, and response inhibition, are engaged in the execution of modified RPS tasks, most studies using this paradigm have mainly paid attention to the frontal areas of the brain to understand underlying neural mechanisms. Consistently, a series of functional magnetic resonance imaging (fMRI) and functional near-infrared spectroscopy (fNIRS) studies that have adopted modified RPS tasks have addressed distinct characteristics of hemodynamics changes in several neural substrates in frontal areas. Paulus and colleagues reported that the inferior prefrontal cortex is highly engaged in computing a trend from previous experiences by adopting the temporal difference model in reinforcement learning, by using RPS games that were specifically modified for the acquisition of advantageous versus disadvantage actions [20]. In addition, by using the same RPS task, these authors also found a pivotal role of the bilateral insula and the medial prefrontal cortex (MPFC) including the anterior cingulate cortex (ACC) in the assessment and action selection stages of decision making [24]. These studies leveraged the usefulness of RPS games for the investigation of decision making and reward processing and found that the frontal areas are predominantly involved in the corresponding cognitive processes. In addition, one fMRI study that used a modified RPS task consisting of three types of goal-directed conditions (draw, lose, and win) reported that the anterior part of the left inferior frontal gyrus is responsible for the inhibition of habitual manual behavior [19]. Two fMRI studies by Kadota and colleagues [18,23] showed the higher activation of the anterior prefrontal cortex in the inhibition of stereotyped responses during a modified RPS task in which subjects were instructed to lose, suppressing the typical response to winning. They also proposed a key role of the dorsolateral prefrontal cortex in the inhibition of stereotyped responses, by assessing blood oxygen level-dependent (BOLD) activation in combination with transcranial magnetic stimulation (TMS) [23]. An fNIRS study also examined hemodynamic changes in the prefrontal cortex (PFC) during a modified RPS task that comprised lose and win conditions [22]. This study revealed increases in hemodynamic activation at specific electrodes corresponding to the left dorsal and ventral lateral PFC (DLPFC and VLPFC), the bilateral premotor area, and the supplementary motor area (SMA). The authors suggested that these linear trends of hemodynamic activation in correspondence with task condition showed the role of the lateral PFC in executive behavioral control related to workload. A recent psychiatric study proposed that the fNIRS signals acquired during a modified RPS task might provide useful information for the evaluation of the cognitive functions of patients with schizophrenia. When such patients performed a lose-conditioned RPS game, the oxygenated hemoglobin changes in the frontal pole area, the dorsolateral prefrontal regions, and the parietal association area were significantly lower than those when healthy subjects performed the same task [21].

Taken together, in spite of some differences in the precise localization due to different task conditions and measurement techniques, there is no doubt that the frontal areas, including the PFC, ACC, SMA, and premotor area, are strongly responsible for the inhibition of habitual behavior during a modified RPS task. The evidence that the frontal areas are highly engaged in the inhibition of habitual behavior is consistent with the results of other SCR tasks, irrespective of the type of behavior. As for other tasks, it is known that the ACC and the DLPFC are closely related to the execution of conflict monitoring, with possible conflicts being detected and evaluated in the ACC and top-down adjustments being exerted in the DLPFC [7,25,26]. As described earlier, neuroimaging studies using modified RPS tasks have sufficiently demonstrated the distinct neural correlates of the inhibition of habitual behaviors [18,19,20,22,23,24]. Unlike past fMRI studies mentioned above, to the best of our knowledges, there is no EEG study to investigate the cognitive control to inhibit habitual behaviors by adopting the RPS paradigms directly. However, it has been established that cognitive control is closely related to the theta oscillations ranging from 4 to 8 Hz in the frontal regions [27,28]. By using the flanker, Simon, and go/no-go tasks, Nigbur and colleagues proposed that theta power is a marker for cognitive interference that could measure enhanced processing demands [29]. In addition, the author showed that the medial frontal cortex established long-range phase synchronization with visual, motor, and lateral prefrontal cortices as well as an increase in theta power in the case of cognitive control demands in a conflicting situation [30]. By using the Simon and flanker tasks, another EEG-beamforming study investigated the neural source interactions in cognitive interferences and proposed that interference processing and conflict resolution were reflected in a broad theta network comprising the prefrontal region, such as the superior, middle, and inferior frontal gyri, and the left supplementary motor area [31].

However, there are still limitations that need to be addressed by further studies. First, as mentioned above, while many studies have revealed specific spatial and anatomical substrates underlying the inhibition of habitual behaviors, little is known about the temporal characteristics of the neural dynamics underlying these processes. Second, these studies have not identified specific network patterns by means of functional or effective connectivity related to these processes. Finally, past studies using RPS tasks have not analyzed response times for the assessment of behavior response. To the best of our knowledges, to date, no study has shown the distinct temporal characteristics of the neural dynamics, including not only local power activity but also global networks, which are involved in these processes. In this study, we thus aimed to reveal the overall spatiotemporal characteristics of the neural dynamics underlying the inhibition of habitual behaviors that are naturally established over a long period of time. To achieve this, we used spectral analysis on electroencephalography (EEG) signals to identify overall spatiotemporal patterns in local neural activities, and phase synchrony analysis on these multichannel signals to recognize the specific characteristics of global functional networks that reflect the inhibition of habitual behaviors during a modified RPS task.

## 2. Materials and Methods

### 2.1. Participants

Seventeen university students aged 21 to 27 years (mean age = 23.29; standard deviation (STD) = 1.93; 12 males and 5 females) participated in this study. Participants were right-handed and had no history of neurological or psychological disorders; they provided written informed consent and were paid for their participation. The study was approved by the ethics committee of Ulsan National Institute of Science and Technology (UNISTIRB-16-37-C).

### 2.2. Rock–Paper–Scissors Task

As one of the most famous games, the RPS game consists of three types of hand shape that represent a rock, paper, and scissors, respectively. Two or more persons simultaneously present one of their hands in one of three shapes. For the task, as described in Figure 1a, the standard rules apply: rock wins scissors, scissors wins paper, and paper wins rock. Based on the standard rules in RPS games, we adopted a modified RPS task comprising a basic RPS (BRPS) task and two different conditioned RPS tasks (CRPS)—lose-conditioned (LCRPS) and win-conditioned (WCRPS) tasks. That is, the modified RPS task in the present study included three different RPS game conditions: (1) BRPS, (2) LCRPS, and (3) WCRPS. These conditions were conducted separately, with two blocks per condition. A BRPS block consisted of 30 trials, and LCRPS and WCRPS blocks consisted of 45 trials, so all participants performed 240 RPS game trials in total over six blocks. Instead of an opponent who played against the participants, three types of visual stimuli represented the RPS actions corresponding to “ROCK”, “PAPER”, and “SCISSORS”. One of these RPS stimuli was presented at the onset of the auditory action cue and remained on the screen until the end of the task. During a resting period, a black cross marker with a white color background for eye-fixation was presented on the screen. The timeline of the RPS task in the present study is illustrated in Figure 1b. The participants played all RPS games against these RPS stimuli on screen accompanied by a sequence of auditory cues that consisted of two different tones (500 and 1000 Hz) instead of the directive sounds “ROCK”, “PAPER”, and “SCISSORS” used in the real RPS game. A trial began with the first preparatory cue (abbreviated to “1” in this study) and was followed by the second preparatory cue (abbreviated to “2”) and the action cue (abbreviated to “A”). The two preparatory cues were identical low (500 Hz) tones, and the action cue was a high (1000 Hz) tone. All auditory cues were presented for 100 ms, with a 500 ms time interval. All subjects perfectly recognized the specific goal of the RPS condition as well as the common procedures of the RPS task prior to the corresponding task.

In the BRPS, the participants were asked to play a classical RPS game in accordance with the auditory cues naturally, so they simply intended to present one of their hands in one of three shapes at the onset of action cue (“A”). Thus, it is natural that the draw, lose, and win rates in the BRPS would theoretically be expected to be 1/3, and the participant’s RPS action would be in accordance with the onset of the RPS action’s cue as in a classic RPS game. Figure 1a shows the standard RPS rules that was applied in the BRPS. In contrast, both CRPS were modified by specific instructions. In the LCRPS, the participants were instructed to lose all games intentionally; they were thus allowed to perform the suitable action after watching the opponent’s action. However, they were instructed to decide and perform their action within 1 s after the onset of the opponent’s action. Reversely, in the WCRPS, we asked the participant to win all games intentionally, using the same method as in the LCRPS. We, therefore, expected a win rate of 0% and 100% in the LCRPS and WCRPS, respectively. For analytic simplicity, in this study, we defined a success rate in both tasks that was ideally expected to be 100% instead using lose and win rates. The task began with two consecutive BRPS blocks followed by two alternating series of LCRPS and WCRPS blocks. All 240 RPS sequences for the opponents’ actions were created prior to the experiment in a controlled random order to maintain the same probability of RPS realization and transition as 1/3 and 1/9, respectively. That is, all probabilities for the occurrence of RPS actions were 1/3, and all probabilities for the transition from the previous RPS action to the current RPS action were 1/9.

### 2.3. EEG/EMG Recordings

Participants performed all experimental tasks in a sound-attenuated and dimly lit chamber. They were asked to sit on a comfortable chair at a distance of 60 cm from a 24 inch full HD LCD computer monitor (1920 × 1080 resolution). All visual RPS stimuli were presented on a white colored background and measured at 22.5 cm width × 20.5 cm height (visual angle: 21.2° × 19.4°) in the center of the screen. A fixation stimulus consisted of a black cross marker (visual angle: 1.91° × 1.91°) on a white colored background the same size as the screen for the RPS stimuli. A ball-shaped supporter was attached to the table for the fixation of the participant’s hand, and all participants placed their right hand on the supporter with an open palm gesture so that all hand movements began from the same gesture and position. We asked participants to perform their RPS actions as quickly but “lightly” as possible to reduce the impact of muscle activity on EEG signals, and then to return their right hand back to the fixation position. While participants were performing the experimental tasks, we simultaneously recorded EEG and electromyography (EMG) signals from the participants’ heads and wrists, respectively. Thirty-two-channel EEG signals were continuously acquired using the amplifier (Brain Products GmbH, Munich Germany) at a sampling rate of 512 Hz with active electrodes mounted on an elastic head cap in accordance with the 10–20 international system. Impedance of all EEG electrodes was kept below 5 KΩ. Online EEG recordings were made with respect to a left-mastoid reference, with a ground lead placed on the right mastoid. In addition, we collected several kinetic signals related to participants’ hand movements that consisted of eight-channel EMG, a three-channel accelerometer, and three-axis gyroscope signals using a wrist-type EMG device (Myo, Thalmic Labs, Kitchener, ON, Canada) at a sampling rate of 500 Hz via wireless Bluetooth communication to detect the exact onset of all individual RPS actions. All participants’ RPS responses were recorded with a video recorder and then manually documented, offline, by two operators using a double cross-checking approach. Because all experimental procedures were executed in accordance with the sequence of three auditory RPS cues, we recorded not only EEG and EMG but also auditory sounds, captured on an embedded sound card, for the detection of exact event times.

### 2.4. Response Time and Accuracy Rate Analysis

For the reliable measurement of individual response times, we first defined the onset time of RPS action as the starting time of participants’ hand movements for RPS actions. Instead of eight-channel EMG and three-channel accelerometer signals, three gyroscope signals were used to determine the individual onset times because these gyroscope signals were more stable and noise free than other signals. As illustrated in Figure 2, we first defined the objective kinetic signals, K(t), as K(t)=gyroX(t)+gyroZ(t)−2∗gyroY(t), where gyroX(t), gyroY(t), and gyroZ(t), corresponding to the *x*-, *y*-, and *z*-axis gyroscope signals, respectively. Despite variations in amplitude and slope across epochs and participants, the gyroscope signals had the same bipolar pattern at the onset of participants’ hand movements; however, the sign of gyroY(t) was reversed in comparison with gyroX(t) and gyroZ(t). Second, we smoothed K(t) by using a moving average filter with an order of 20 and determined the positive and negative reference points as the first points that exceeded the predefined double threshold set at ±0.1 × STD of K(t). All negative reference points were earlier than the corresponding positive reference points. Finally, the onset point of RPS action was determined by the first point that crossed zero values from the negative reference point (in reverse). The response time was defined as the latency time between the onset of the action cue (A) and the onset of the hand movement obtained from the gyroscope signals. For the assessment of the behavioral data, we quantified two measurements, response time (RT) and success rate, to evaluate the participants’ performance of the RPS tasks and then examined these measurements statistically using two-way repeated-measures ANOVAs (rmANOVAs) with the within-subject factors order (first/second) and task (LCRPS/WCRPS). False-discovery rate (FDR) corrections for multiple comparison were applied [32].

### 2.5. EEG Preprocessing

All EEG and EMG analyses in this study were conducted off-line using the EEGLAB toolbox [33] and custom Matlab codes (Mathworks, Natick, MA, USA). The EEG signals were first band-pass filtered from 0.1 to 55 Hz using a zero-phase finite impulse response (FIR) filter. To eliminate eye movement and muscle artifacts, we performed an independent component analysis, removed the corresponding components manually by visual inspection, and then recomposed them into the EEG signals. We constructed EEG epochs covering the entire RPS task period from 1 s before the onset of the first preparatory cue to 1 s after the action cue, with an additional residual margin (±1 s) to avoid unnecessary edge artifacts from further signal processing analysis. The start and end points of EEG epochs were abbreviated as “S” and “E”, respectively.

### 2.6. EEG Spectral Analysis

Prior to the following spectral and phase synchrony analyses, we first removed “bad” EEG epochs by checking whether a trial represented a failed mission or whether the corresponding RT exceeded 1 s after the onset of the action cue. Only the remaining “valid” EEG epochs were used for further statistical analyses. From the 90 trials in both CRPS, the average and STD of the number of valid EEG epochs across all participants were 77 ± 15.7 for the LCRPS and 89 ± 9.0 for the WCRPS, respectively.

Next, for the overall spectral characteristics of induced oscillations during the task, a spectral analysis based on continuous wavelet transform estimated the event-related spectral perturbation from the time-series EEG signal. The EEG signal was first convolved with a complex Morlet wavelet: $w\left(t,f_{0}\right)={\left(\sigma_{t}\sqrt{\pi }\right)}^{-1/2}e^{-t^{2}/2\sigma_{t}^{2}}e^{2\pi if_{0}t}$, where i was the imaginary unit, $\sigma_{t}=m/2\pi f_{0}$. For the trade-off between time and frequency resolution, we determined the constant m = 5, in line with the study conducted by Nigbur and colleagues [29], yielding a time and frequency resolution of the theta frequency band at $f_{0}$ = 6 Hz of $\sigma_{t}$ = 0.133 ms and $\sigma_{f}$ = 1.2 Hz, respectively. In each trial, spectrograms ranging from 2 to 55 Hz were generated by averaging the squared absolute values of the convoluted values. The boundaries for the baseline correction ranged from −0.5 to −0.1 s after the onset of the first preparatory RPS cue. The power values for each frequency were applied to a log-transform and subtracted from the mean of the baseline power. All spectrograms in an epoch were ordered by condition (BRPS/LCPRS/WCRPS) and by task (first/second for each condition) for the following statistical analyses. To remove the unwanted edge effect from signal processing, all spectral procedures were performed in the total period of EEG epochs. The residual margins (±1 s) were then eliminated from the EEG epochs.

### 2.7. EEG Phase Synchrony

Because we found a significant difference in the spectral power of theta frequency (4–8 Hz) oscillations among the RPS conditions exclusively, these modulatory oscillations were intensively examined with respect to the distinct characteristics of their neural synchrony. More detailed spectral results are described in the Results section below. To quantify the spatiotemporal characteristics of functional connectivity in theta oscillations during the tasks, we estimated all phase locking values (PLVs) in each pair of EEG channels by performing a phase synchrony analysis [34,35]. For the calculation of PLVs, we first extracted instantaneous phase values of all EEG signals by applying the Hilbert transform in combination with a band-pass filter as follows. The EEG signal was filtered in the theta frequency range of 4 to 8 Hz using an FIR filter with an order of 256. If the filtered signal is defined as x(t), by using the Hilbert transform, we can determine the analytic signal, $z\left(t\right)=\;x\left(t\right)+i\hat{x}\left(t\right)$. The imagery part of $\hat{x}\left(t\right)$ is the Hilbert transform of x(t) defined as $\hat{x}\left(t\right)=\frac{1}{\pi }\bar{P}\int_{-\infty }^{\infty }\frac{x\left(\tau \right)}{t-\tau }d\tau$, where $\bar{P}$ is the Cauchy principal value operator. By using the analytic signals expressed as $z\left(t\right)=Ae^{j\left(\omega t+\theta \right)}$, the instantaneous phase value at time t was defined as $\theta \left(t\right)=arctan\left(\frac{\hat{x}\left(t\right)}{x\left(t\right)}\right)$, where A is the amplitude of signal x at time t, ω is the frequency, and θ is the phase. All filtering procedures for phase synchrony analyses were performed with a two-way, zero phase-lag, least-squares FIR filter to avoid phase distortion. To remove artifacts from edge effect during the signal processing, all filtering procedures were performed in the continuous EEG signals. After converting the EEG epochs from these continuous theta-filtered EEG signals, the residual parts of the EEG epochs were eliminated as in the spectral analysis in this study. Next, we calculated instantaneous phase differences of all pairs by subtracting the calculated instantaneous phase values of the corresponding two signals. As we defined the PLVs between two signals as the degree of consistency in phase difference between them at the same time across trials, these values were calculated as the average of instantaneous phase differences at the same time across trials. Therefore, PLVs ranged from 0 to 1; that is, if two signals were random, this value was close to 0; otherwise, if two signals were perfectly synchronized, this value was close to 1.

### 2.8. Statistical Evaluation

The PLVs for all task conditions were statistically analyzed to determine whether a pair of EEG sites was significantly synchronized or desynchronized by using the permutation test, called phase locking statistics, proposed by Lachaux and colleagues [35,36] and replicated by Trujillo and colleagues [37]. Specifically, sample by sample, in each RPS task condition, we constructed a set of surrogate PLVs in which the trial order at one EEG site was randomly shuffled, and the PLVs between two sites were repeatedly calculated. A sample in an RPS task condition had one original PLV and 500 surrogate PLVs because 500 permutation tests were conducted. Time courses of both the original PLV and the 500 surrogate PLVs were individually normalized by subtracting the mean and by dividing the STD of the baseline interval at −1.4–−1.0 s before the onset of the RPS action cue. By averaging across 500 surrogate PLVs already transformed to z-score PLVs, we created the representative surrogate z-scored PLVs for each RPS task condition. To display the spatiotemporal patterns of synchronized theta pairs, we divided the entire time course PLVs into eleven 200 ms time windows, as in the spectral analysis. For each time window and each RPS condition, paired t-test analyses were used to compare the original PLV and a distribution of averaged surrogate PLVs to decide whether the original PLV was significantly higher or lower than the mean of the surrogate PLVs in each time window (*p* < 0.05; false-discovery rate (FDR)-corrected). Finally, the statistical results obtained from all participants in the same time window were aggregated, and nonparametric paired Wilcoxon tests were conducted to determine a significant increase or decrease in synchrony for each individual time window in comparison with the baseline (*p* < 0.05; FDR-corrected).

## 3. Results

### 3.1. Response Times and Success Rates

For the RT analyses, we compared the differences in RTs between the BRPS and the CRPS, to clarify how the different RPS task conditions affect RTs. RT data were separately aggregated and represented as frequency histograms with gaussian fitting, as illustrated in Figure 3a. The average and STD were −0.5 ± 0.172 s for the BRPS and 0.245 ± 0.155 s for the CRPS (where movement onset was set as 0 s), respectively. Note that the RT defined in this study was the immediate time of subjects’ hand movements detected by the kinetic signals. So, the mean RT in the BRPS was faster than the time of action cue, because most subjects started their actions before the onset time of the auditory action cue in the BRPS. A two-way rmANOVA on RTs revealed a faster performance for the first block (F(1,16) = 39.361; *p* = 0.000) and a faster performance for the WCRPS (F(1,16) = 10.745; *p* = 0.005). There was no interaction effect (F(1,16) = 0.040; *p* = 0.844). The RT average and STD were 0.22 ± 0.10 sec for the first LCRPS, 0.31 ± 0.08 s for the second LCRPS, 0.18 ± 0.10 s for the first WCRPS, and 0.27 ± 0.07 s for the second WCRPS (Figure 3b), respectively.

For the success rate analyses, the rmANOVA for success rates revealed higher success for the second block than for the first block (F(1,16) = 9.729; *p* = 0.007) and higher success for the WCRPS than for the LCRPS (F(1,16) = 7.948; *p* = 0.012) (Figure 3c). There was no interaction effect (F(1,16) = 1.628; *p* = 0.220). The average and STD of the success rate increased from 0.85 ± 0.20 for the first LCRPS to 0.90 ± 0.17 for the second LCRPS, and from 0.95 ± 0.13 for the first WCRPS to 0.97 ± 0.07 for the second WCRPS, respectively. These results showed that there was both a repetition effect and lower task performance in the LCRPS compared to the WCRPS. Concomitantly, we could obtain the win, draw, and lose rates instead of the success rate in the BRPS, because there was no goal-directed behavior. In the BRPS, the average and STD were 0.343 ± 0.084 for the lose rate, 0.329 ± 0.058 for the draw rate, and 0.333 ± 0.064 for the win rate, respectively. As expected, there was no statistical difference among them. Regarding these behavior analysis results, we can summarize that the participants were more careful to respond with the proper RPS action by accepting a slightly slower RT in the second block compared to the first block. Moreover, they found that it was more difficult to perform the LCPRS compared to the WCPRS, so it took longer to execute the proper RPS action in the LCRPS.

### 3.2. Overall Spectral Analysis

To examine the overall spatiotemporal characteristics of the induced EEG oscillations, we constructed power spectrograms ranging from 4 to 50 Hz within an EEG epoch for each RPS task ORDER and CONDITION, respectively. By performing two-way rmANOVAs with 2 (ORDER: first/second) × 3 (CONDITION: BRPS/LCRPS/WCRPS) factors, we examined statistical differences in spectral power among the three RPS task conditions and found no main effect of ORDER and no interaction effect; we did, however, find a main effect of CONDITION. Figure 4a, for example, illustrates three power spectrograms calculated from the Fz channel that belongs to the mid-frontal area corresponding to BRPS, LCRPS, and WCRPS, respectively. As shown in these logarithmic power spectrograms before baseline correction, the three RPS task conditions consistently modulated both theta (4–8 Hz) and alpha (8–13 Hz) bands but no other frequency bands. That is, the RPS action onset triggered the pronounced decrease in alpha power activities and increase in theta power activities. Moreover, as shown by the statistical results on the main effect of CONDITION in Figure 4b, statistical differences in spectral power among the three RPS tasks were exclusively observed in theta oscillations from 4 to 8 Hz in the mid-frontocentral regions (FPz, F3, Fz, FC1, FC2, C3, and Cz). As with the other frequency bands, the power decreased in the alpha band after the onset of RPS actions, which did not show any significant difference among conditions.

### 3.3. Spectral Analysis of the Induced Theta Oscillations

Based on the observation of overall spectral changes during the RPS tasks illustrated in Figure 4, we only focused on the induced theta oscillations and examined their specific spatiotemporal characteristics specifically. First, the time course for theta power signals was built by extracting and aggregating the spectral power corresponding to the theta band. Next, for the significant differences in induced theta power between the LCRPS and WCRPS, we conducted paired t-tests to examine spatial and temporal differences. The analyses revealed that the statistical differences between the LCRPS and WCRPS were significant at four sites (FPz, F3, Fz, and FC2) in the frontocentral regions between 0.25 and 0.50 s (Figure 5a). Within this time interval of interest, Figure 5a shows the 3D spatial distribution of theta power across all scalp regions in the LCRPS (left top) and the WCRPS (right top). Two 2D spatial distributions are illustrated as power differences in theta activity between both conditions and corresponding t-values in the left bottom panel and the right bottom panel of Figure 5a, respectively. Figure 5b shows the three individual theta power time course signals calculated by averaging over the four significant sites corresponding to the BRPS, LCRPS, and WCRPS. Interestingly, these time course signals showed both common and distinct temporal characteristics of theta activities modulated by a sequence of RPS tasks. Specifically, theta powers fluctuated according to two cues, a preparatory and an action cue, and two or three peaks of power activity were observed at the onset time of these cues, with some time delay. In the BRPS, three peaks of theta power time course signals appeared to be observed, in accordance with the onset of the preparatory and the action cue (1, 2, and A). The enhancement in theta power occurred consistently for the interval between the two preparatory cues and disappeared after the onset of the action cue (0 s). In contrast, in both CRPS, there were two large peaks of increased theta power corresponding to the onset of the first preparatory cue and the action cue (1 and A), while the mid-peak of theta activity corresponding to the second preparatory cue (2) markedly decreased and disappeared in both tasks.

### 3.4. Phase Synchrony Analysis of the Theta Oscillations

For the examination of the spatiotemporal characteristics of phase synchrony in theta oscillations, we first divided the sample-by-sample statistical results of the PLV validation test into eleven 200 ms intervals and then constructed the individual adjacent matrix representing which pairs were significantly synchronized or desynchronized. Figure 6a illustrates the three types of topographical patterns of significant synchronized pairs in the theta oscillations induced by the different task conditions. As shown in Figure 6a, the strongest phase synchronization in theta oscillations was observed in the LCRPS between 0 and 600 ms after the onset of the RPS action cue. As for other conditions, significant phase synchronization pairs occurred rarely in the BRPS, and only a few synchronized pairs were observed in the WCRPS. To evaluate the binary networks in all individual adjacent matrices, we calculated two measures, the degree of nodes and efficiency. Figure 6b,c show the degree of nodes and efficiency for all RPS conditions, respectively. Both measures showed the early presence of synchronized pairs in the WCRPS and the stronger and more pronounced synchronization in the LCRPS.

Based on these temporal patterns of power activation and of phase synchronization in theta oscillations during the task, we further examined their occurrence in time. For the examination of time differences between power activation and phase synchronization, the relevant signals corresponding to all RPS task conditions needed to be defined at the same time by reorganizing all corresponding time course signals. For this purpose, we averaged all time course signals of frontal theta power amplitudes, which were obtained from the Fz channel, and log-transformed and normalized all power values. We constructed two representative time-course signals of power amplitude corresponding to the LCRPS and the WCRPS for each trial and each subject within a 200 ms time window with a 50 ms shift time (25% overlap). Figure 7a shows the time-course numbers of synchronized pairs in the LCRPS and WCRPS separately. These results show the early presence of synchronized pairs in the WCRPS and the stronger and more pronounced synchronization in the LCRPS. Figure 7b illustrates the mean and standard error signals of theta power amplitudes for both CRPS. In addition, we obtained peak times of theta power activities for these time-course signals from all subjects. Figure 7c shows the peak time distributions of theta power for both CRPS (first two boxplots) and the time distribution of the maximum differences between these within an individual (last boxplot). The mean and STD of the RPS trial peak times were 0.426 ± 0.229 s in the LCRPS and 0.352 ± 0.240 s in the WCRPS, respectively, and these signals were maximally different at 0.526 ± 0.194 s. For the representation of time-course signals of the degree of theta phase synchronization, we performed an additional phase synchrony analysis (as described in the Methods above) by matching the time resolution that was the same as that of the theta power amplitude calculation shown in Figure 6b, with a 200 ms time window and a 50 ms time shift (25% overlap). For each time window, we calculated the total number of synchronized pairs corresponding to the LCRPS and WCRPS, respectively. Because we defined synchronized connections across all subjects (as described in the Methods), we obtained the total time-course signals for the total number of paired connections with peak times (marked with red and blue squares for the LCRPS and WCRPS, respectively, in Figure 6a for each condition). To examine the significant differences in power activation times and phase synchronization between the two CRPS, we conducted one-sample Wilcoxon signed rank tests to determine whether the peak time of phase synchronization was statistically different from the mean of the distribution of peak times. These analyses revealed that the peak times of phase synchronization in both the LCRPS and the WCRPS came before the power activation in each condition (T = 84, Z = 2.665, *p* < 0.005 for the LCRPS; T = 153, Z = 3.602, *p* < 0.001 for the WCRPS; T is the value of the sign rank test that indicates the sum of the ranks of positive differences, and Z is the value of the z-statistics). In addition, these analyses showed that the time of the maximum difference in phase synchronization between the LCRPS and the WCRPS preceded the time of maximum differences in power activation (T = 133, Z = 3.343, *p* < 0.001). In sum, these analyses revealed two main findings related to the time difference between two different neural dynamics, namely, local power and global networks. First, in both the LCRPS and the WCRPS, the activation of the global network preceded the activation of local power. Second, the differences in phase synchronization between the LCRPS and the WCRPS appeared before differences in power activation.

## 4. Discussion

Grounded in the previous findings on neural correlates of the inhibition of habitual responses from past neuroimaging studies, the present study extended these analysis to the spatiotemporal characteristics of neural dynamics underlying this cognitive process. In the following, we discuss the behavior and neural mechanisms underlying cognitive control related to long-term established, habitual behavior responses established involuntarily without any effortful learning.

### 4.1. Temporal Variation in Frontal Theta Power Reflects a Stream of Cognitive Processes

Considering the special role of the frontal area in cognitive control, it is not surprising that conducting the modified RPS used here strongly reflected the neural dynamics over the frontal regions specifically. For decades, many neuroimaging studies have addressed the observation that the frontal regions in the brain, including the PFC, ACC, medial frontal cortex (MFC), and SMA, are highly involved in the processing of cognitive control [38,39,40]. One meta-analysis of human fMRI studies concluded that the medial frontal cortex is engaged in cognitive control related to adaptive goal-directed behavior that is needed to monitor and adjust ongoing performance [26]. However, new findings in our study indicated that the theta power amplitude was significantly higher in the LCRPS than in the WCRPS and that the temporal characteristics of theta power in the frontal areas directly reflected the stream of cognitive processes involved in the corresponding RPS task. In the BRPS, which lacked any instructions, the time-course theta power showed three distinct peaks with similar amplitudes in accordance with the three auditory cues for the RPS game. In contrast, in both the LCRPS and the WCRPS, which included specific goal-directed instructions, there were only two peaks of theta activity, corresponding to the first preparatory and the action cues. Given the waning and waxing of frontal theta power, we addressed that this apparently revealed the typical temporal dynamics of theta power as a mediator of cognitive processes, although S-R mappings reflected customary and habitual behaviors that were established without any explicit training. Considering this close relationship between frontal theta power and cognitive processes, we speculate that the theta power peaks reflect the distinct stages of cognitive processes, such as preparation and action for performing the corresponding RPS tasks, and postulate that the differences in cognitive control between the RPS tasks induce differences in the neural dynamics underlying theta oscillations sourced from the frontal areas, including the PFC and the ACC.

### 4.2. Activation of Phase Synchronization Precedes Power Activity

One main finding in this study is that stronger phase synchronization was observed in the LCRPS than in the WCRPS. We proposed that this appeared to reflect the processing of cognitive control in which it largely engages in global interactions for inter-regional communication. Generally, the functional characteristics of phase synchronization are known to play pivotal roles in neural communication and plasticity [41]. Much evidence supports that a variety of executive functions require specific phase-synchronized networks among task-relevant brain regions. Among them, the phase synchronization in the theta oscillations is among the most important media in decision making [42,43], visual perception [44], working memory [41], goal-directed behavior [45], and conflict detection and resolution processes [46]. Like the habitual behavior responses in this study, Cavanagh and colleagues reported that error-related EEG activities generated strong phase synchronization between the medial (FCz) and lateral (F5/6) PFC when the participants performed action monitoring in a modified Eriksen Flanker task. The authors addressed that this finding reflected a mechanism of communication between networks related to action monitoring and cognitive control [47]. Similarly, using the speeded flanker tasks for perceptual and response conflicts, Nigbur and colleagues observed that theta phase synchronization between the medial frontal (FCz) and lateral frontal (F5/6) sites w only as enhanced at the occurrence of response conflict in the incongruent response. The authors suggested that theta oscillations in the MFC were engaged in conflict processing and linked with a neural network for control response conflict [30]. In line with these previous findings on the phase synchronization in the theta oscillations related to cognitive control, our results demonstrated that the higher cognitive control to resolve the LCRPS condition requires large-scale neural communication between task-related regions, which strongly modulates the dynamics of theta oscillations. In addition, Cohen and Donner investigated the dynamics of theta oscillations in association with action monitoring and conflict resolution in the Simon and auditory–visual Simon tasks [46]. The authors divided the properties of ERP signals into phase-locked and nonphase-locked components and proposed that the power of nonphase-locked theta-band oscillations within the MFC is more reliably modulated by the cognitive control rather than by phase-locked EEG components. In this study, we found that the onset time of power activation in the phase-locked components was faster than that in the nonphase components. Regarding the onset time difference between phase synchronization and power activation in the LCRPS, we proposed that the precedent theta phase synchronization reflected in the large-scale neural communication within the brain before the proper task-relevant regions mainly operated in specific goal-directed cognitive controls.

### 4.3. The LCRPS Is More Difficult Than the WCRPS

Statistical analyses on the behavioral data showed that it was more difficult for the participants to perform the LCRPS than the WCRPS. Generally, this finding is line with the evidence shown in two past studies using modified RPS tasks [19,23]. In the Matsubara study, during the modified RPS tasks consisting of DRAW, LOSE, and WIN conditions in the right-hand block, the error rate was higher for the LOSE condition than for the WIN condition. Given these behavior results, the authors suggested that it is natural behavior for people to attempt to win the game [19]. In the Kadota study, a significant difference in the mean correct performance was observed between two different groups, a win group and a lose group. The win group achieved higher correct rates than the lose group [23]. Consequently, our behavior results were in line with both intersubject and intrasubject evidence that the participants were accustomed to trying to win the game. In addition, these behavior characteristics were firmly supported by the statistical analysis on the response time, which has not been addressed before. Notably, we first defined the response time as the exact onset of hand movement in this study. To assess and analyze the RT data, it is necessary to define a fiducial time point as the response time during the real hand movement in the task. Therefore, most of the RTs in the BRPS preceded the onset of the action cue because all participants were just starting to move their hand before they heard the action cue. Given the behavior results of RTs showing a slower RT for the LCRPS compared to the WCRPS, we verified that it is more difficult to perform the LCRPS, which demands more cognitive processing. Similarly, a recent study by Cooper and colleagues reported that the increase in frontal theta power could predict the size of the cognitive switch cost and decreases in the RT [48]. Therefore, we concluded that the difficulty in the LCRPS led to lower success rates and longer RTs in the LCRPS, which required higher cognitive control to inhibit habitual responses.

### 4.4. Limitations and Future Work

The present study demonstrates the pivotal role of the frontal regions in the inhibition of habitual behavior by showing distinct temporal patterns in theta dynamics. However, it has several limitations. First, due to the small number of EEG channels used in this study, we could not perform source localization to investigate specific neural correlates of these processes. There are still some controversial arguments about how many EEG channels are needed for reliable source localization. The 32 EEG channels used in this study barely satisfy the minimum number for reliable source localization [49,50], so we could not guarantee that some results from source localization provided more reliable findings compared with past anatomical results from fMRI studies. Second, we were not able to account for the observed repetition effect using long-term training on the LCRPS because there were only 90 trials (2 blocks) of the LCRPS performed in one day. With more trials, we may address the following questions: Is it possible that the inhibition of habitual behavior can be trained? If it is possible, how does the difference in RPS tasks change the response times and spatiotemporal characteristics of neural dynamics? We speculate that it would be possible to increase the performance of the LCRPS to close to the level of the WCRPS through training, and the differences in neural dynamics between both CRPS tasks would be reduced. However, this should be verified and explained by further studies.

## 5. Conclusions

This study identified that the cognitive control for the inhibition of habitual behaviors intensively modulated the spatiotemporal changes in theta oscillations. In line with the degree of difficulty in this cognitive control, the preceding global phase-synchronized network and the following frontal power were activated in the theta oscillations. These results demonstrated that the cognitive inhibition of habitual behaviors was apparently reflected in the theta oscillations, although it was established involuntarily without any effortful learning.

## Figures and Tables

**Figure 1 brainsci-11-00368-f001:**
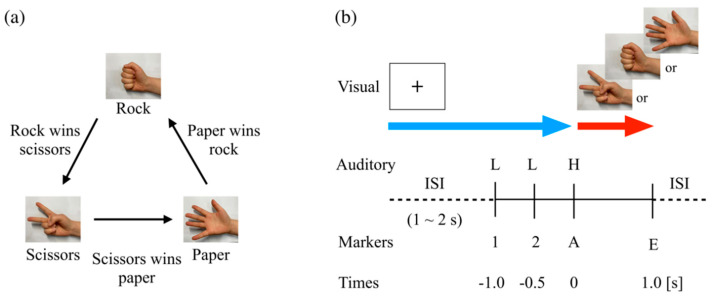
Diagram of RPS task procedures. (**a**) Standard RPS rules. (**b**) In the visual stimulus, a cross marker for eye-fixation and three types of visual stimuli for each RPS action were presented on the screen. The blue arrow indicates the duration of presentation of the fixation stimulus. The red arrow indicates the duration of presentation of one of the RPS actions. The interstimulus interval (ISI) varied from 1 to 2 s. “1” and “2” indicate the first and second preparatory cues, respectively. “A” is the action cue occurring at 0. “E” indicates the end of the trial at 1.0 s.

**Figure 2 brainsci-11-00368-f002:**
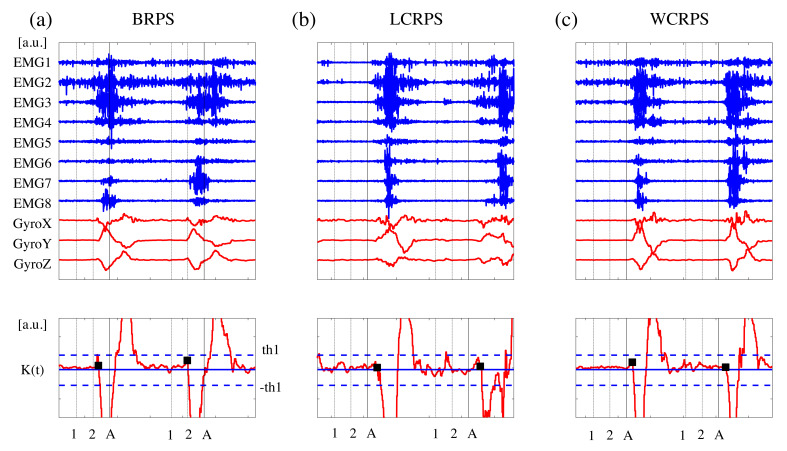
Examples of electromyography (EMG) and kinetic signals and illustration of the onset detection algorithm. (**a**) basic rock–paper–scissors RSP (BRPS), (**b**) lose-conditioned RPS (LCRPS), (**c**) win-conditioned RPS (WCRPS), (top) eight EMG signals (blue color), three gyroscope signals (red color). (bottom) Kinetic signals (red color) for the onset detection of participants’ hand movements. The black cross indicates the detected onset time. The two blue broken lines indicate the positive and negative thresholds for the reference points, respectively. The black vertical lines indicate the time points in the RPS epoch. The two dotted lines correspond to the two preparatory cues, and the solid line corresponds to the action cues.

**Figure 3 brainsci-11-00368-f003:**
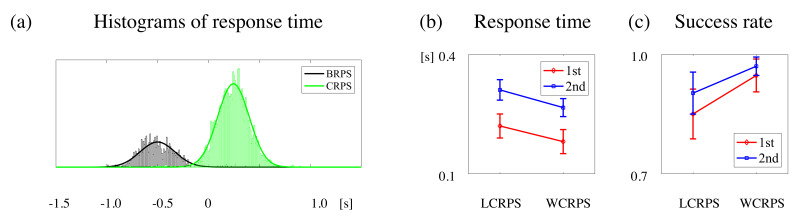
Statistical results of behavior data. (**a**) Two distributions of response times to RPS action in BRPS (black color) and CRPS (green color), respectively. (**b**) Response times for the LCRPS and WCRPS for the first (red) and second (blue) sessions, respectively. (**c**) Success rates for the LCRPS and WCRPS for the first (red) and second (blue) sessions, respectively.

**Figure 4 brainsci-11-00368-f004:**
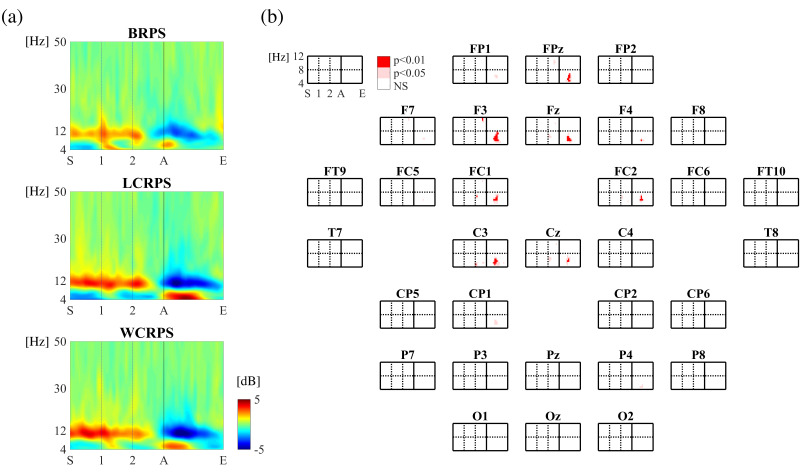
Overall spectrograms for the three RPS tasks. (**a**) Three spectrograms of overall spectral characteristics obtained from the Fz channel corresponding to BRPS (top), LCRPS (middle), and WCRPS (bottom), respectively. (**b**) Topographies of statistical results on power differences among the three RPS tasks (one-way ANOVAs); statistical differences are indicated by light red (*p* < 0.05) and dark red (*p* < 0.01) colors (FDR-corrected). In both A and B, “S” indicates the beginning of an individual trial at −1.5 s, and “1” and “2” indicate the first and second preparatory cues at −1.0 and −0.5 s, respectively. “A” is the action cue occurring at 0. “E” indicates the end of the trial at 1.0 s.

**Figure 5 brainsci-11-00368-f005:**
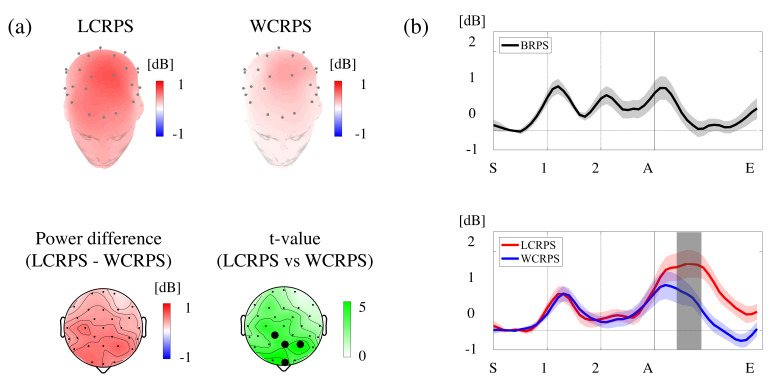
Spatiotemporal characteristics of theta power and phase synchronization with statistical results for all tasks. (**a**) Two 3D topographies of the theta power in the LCRPS (top left) and the WCRPS (top right), respectively, and two 2D topographies of the power difference between the LCRPS and the WCRPS (bottom left), respectively, as well as the corresponding t-values; four black circles in the frontocentral areas (FPz, F3, Fz, FC2) indicate statistical significance (*p* < 0.05, FDR-corrected) (bottom right). (**b**) Time course theta powers averaged across the four EEG sites belonging to the frontocentral regions. The dark area indicates the time interval of statistical differences (*p* < 0.01, FDR-corrected). “S” indicates the beginning of an individual trial at −1.5 s, and “1” and “2” indicate the first and second preparatory cues at −1.0 and −0.5 s, respectively. “A” is the action cue occurring at 0. “E” indicates the end of the trial at 1.0 s.

**Figure 6 brainsci-11-00368-f006:**
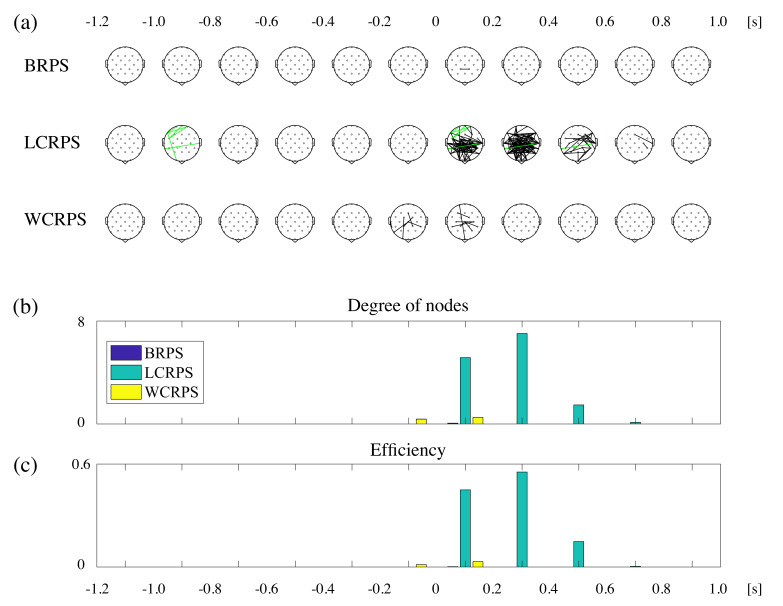
Spatiotemporal characteristics of theta phase synchronization for all RPS tasks. (**a**) Three series of topographies for spatiotemporal patterns in theta phase synchronization during each RPS task. The black lines mean synchronized pairs, and the green lines mean desynchronized pairs in comparison with the baseline (*p* < 0.05, FDR-corrected). The time stamps in the top indicate the corresponding time interval. (**b**) Bar graph for the degree of nodes. (**c**) Bar graph for efficiency. In (**b**,**c**), the time stamps at the bottom indicate the corresponding time interval.

**Figure 7 brainsci-11-00368-f007:**
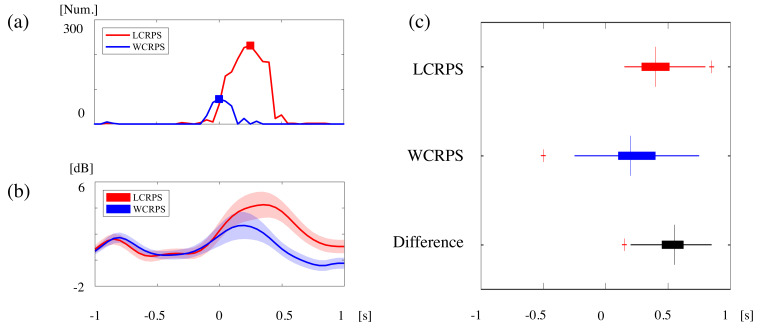
Time differences between power activation and phase synchronization in theta oscillations during the tasks. (**a**) Time course signals of the number of paired connections in the phase synchronization network in the LCRPS (red) and the WCRPS (blue), respectively. Red and blue square markers indicate the peak times, respectively. (**b**) Time course signals of the average and standard error of the theta power in the LCRPS (red) and the WCRPS (blue), respectively. (**c**) Boxplots for the distribution of individual peak times of theta power in the LCRPS (red) and the WCRPS (blue), respectively, and time differences between the two conditions (black). As in A, red and blue square markers indicate the peak times of phase synchronization in the LCRPS and WCRPS, respectively, and the black square marker is the peak time of time difference between them.

## Data Availability

Data available on request due to restrictions eg privacy or ethical.

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
