# Peer review of "Spatiotemporal Characteristics of Neural Dynamics in Theta Oscillations Related to the Inhibition of Habitual Behavior"

_brainsci, 2021, doi:10.3390/brainsci11030368_

Round 1

Reviewer 1 Report

The manuscript by Kang et al. described a study based on a modified rock-paper-scissors (RPS) task with EEG. They applied spectral and phase synchrony analyses in the data and found that, compared to win-conditioned RSP (WCRPS), participants in lose-conditioned RSP (LCRPS) showed slower response time and increased frontal theta power and theta oscillation phase synchrony, which could be an indication of cognitive control. The manuscript is in general well-written and the findings are interesting. I have several comments as follows:

Major:

  1. The introduction lacks critical literature reviews in EEG data on behavioral inhibition/conflict resolution, which seems to be the focus of this manuscript. The study should be theoretically motivated.
  2. The authors may need to be consistent about whether to include BRPS in the reporting. Currently, while they included the BRPS condition in the spectral analysis, the RT and accuracy information was not reported.
  3. The difference in theta oscillations between LCRPS and WCRPS conditions seems to be related to a difference in how long the participants took to respond to these conditions. Can differences in RT be effectively controlled to show more fundamental cognitive effects (e.g. align epochs to the response)?
  4. An increase in phase synchrony can be caused by overall more activity. How can the authors disentangle these two?
  5. The authors explained increased theta activity as more cognitive control, cognitive inhibition, behavior inhibition, or increased cognitive load. These are all different concepts and the explanations are largely speculative. How can the authors find support in the current data to support these claims?
  6. Were trials included in the analyses correct trials only or all trials?

Minor:

  1. The authors used the word “highly” many times, e.g. as in “highly modulate” or highly reflected”. I recommend changing to more appropriate wording
  2. Some figures are not very legible (e.g. the vertical lines in Fig 4B)
  3. Not sure if the last section – patents is intended.

Author Response

We deeply appreciate your questions and comments. According to the reviewers’ comments, we mainly revised the introduction part as well as other parts. Details of our responses to every comment are present below. Please refer the page and line in the revised manuscript with "_trackchanged" version not "_clean" version.

Reviewer 2 Report

Undoubtedly, the human brain carries out cognitive control for the inhibition of habitual behaviors by suppressing some familiar but inappropriate behaviors instead of engaging specific goal-directed behavior flexibly in a given situation. To examine characteristics of neural dynamics related to such inhibition of habitual behaviors, Authors used a modified rock-paper-scissors task that consisted of a basic, a lose-, and a win-conditioned game. 

My comments to the study are as follows:

1. As part of the introduction, I propose to refer to the EEG signal, electroencephalography, and fMRI methods more broadly. For this purpose, I propose a name related to, among others positions: Data Acquisition Methods for Human Brain Activity, Analysis and classification of eeg signals for brain-computer interfaces, Studies in Computational Intelligence from 2020. This will also have a positive impact on updating the bibliography with new literature items from the latest years.

2. The graphs in Fig. 2 and Fig. 3 should have an exact description of the axes.

3. The data contained in Fig. 4B is difficult to read. Data presentation in this respect should be improved.

4. The description of Fig. 5 is too long for its name. The content there should be part of the article section in the form of a paragraph with reference to Fig. 5.

5. I propose to extend the Conclusions with plans for the future. This context is not there. I believe that point 4.4 should be part of Conlusions.

Author Response

(The authors gave the same response as above.)

Reviewer 3 Report

This study attempted to examine the neural dynamics of theta oscillations putatively associated with the inhibition of habitual behavior.   The study is intended to investigate the temporal characteristics of neural dynamics associated with the inhibitory control of behavior using theta oscillations and measures of EEG phase synchrony. In the Introduction the authors vaguely refer to a children's game to be used as an experimental paradigm, based on some prior fMRI and fNIRS studies using a similar task model. However, there is no clear theoretical framework to address this problem, nor any hypotheses being examined. Moreover, there are serious methodological limitations that compromise a sound interpretation of the  behavioral and EEG results of the study, as described in the following list of main concerns:   Main concerns:  

  1. In the Methods section, the task is insufficiently described. For instance, it is not clear what participants had to respond and when (i.e., upon display of one of the three visual stimuli? and what was the response window allowed?)
  2. There is no theoretical justification for the two auditory cues presented in anticipation of the visual stimulus. Why were these cues presented at all? And why were there two low tones and one high tone?
  3. Likewise, it is not clear how the two "control" conditions, "all lose" and "all win" trials, were implemented. For this participants were "allowed to perform their action after the opponent's action" (p. 3). But how was this option implemented in the task?  This instruction must have involved different response windows for the different task conditions, which introduced strong response biases and  distinct speed-accuracy tradeoffs across task conditions that are not accounted for in the Method section.
  4. All those response biases surely influenced behavioral and brain responses in a unknowable manner. And, for the same reasons, it is not possible to discern what aspects of those tasks captured "the inhibition of habitual behavior".
  5. The poor quality of some Figures mean that some behavioral and EEG results are barely visible (e.g., Figure 3, Figure 4B, Figure 5C).

  In the light of the conceptual and methodological shortcomings described above, the discussion of results and conclusions of the study would need a complete reappraisal.   

Author Response

(The authors gave the same response as above.)

Round 2

Reviewer 3 Report

The authors have adequately addressed all of my previous concerns in their revised MS. Now I believe this study makes a nice contribution to the purported role of theta oscillations during the inhibition of habitual behavior.